# A Realist Evaluation of Team Interventions in Acute Hospital Contexts—Use of Two Case Studies to Test Initial Programme Theories

**DOI:** 10.3390/ijerph18168604

**Published:** 2021-08-14

**Authors:** Una Cunningham, Aoife De Brún, Mayumi Willgerodt, Erin Abu-Rish Blakeney, Eilish McAuliffe

**Affiliations:** 1Centre for Interdisciplinary Research, Education and Innovation in Health Systems (UCD IRIS), School of Nursing, Midwifery & Health Systems, University College Dublin, D04 V1W8 Dublin 4, Ireland; aoife.debrun@ucd.ie (A.D.B.); eilish.mcauliffe@ucd.ie (E.M.); 2Pillar Centre for Transformative Healthcare, Mater Misericordiae University Hospital, Eccles St, D07 R2WY Dublin 7, Ireland; 3Department of Biobehavioral Nursing and Health Informatics, School of Nursing, University of Washington, Seattle, WA 98195, USA; mayumi@uw.edu (M.W.); erin2@uw.edu (E.A.-R.B.)

**Keywords:** team, interventions, quality and safety, acute hospital, contexts, realist evaluation, generative causation, initial programme theories, middle-range theories, case studies

## Abstract

Background: Designing and implementing team interventions to improve quality and safety of care in acute hospital contexts is challenging. There is little emphasis in the literature on how contextual conditions impact interventions or how specific active ingredients of interventions impact on team members’ reasoning and enact change. This realist evaluation helps to deepen the understanding of the enablers and barriers for effective team interventions in these contexts. Methods: Five previously developed initial programme theories were tested using case studies from two diverse hospital contexts. Data were collected from theory driven interviews (*n* = 19) in an Irish context and from previously conducted evaluative interviews (*n* = 16) in a US context. Data were explored to unpack the underlying social and psychological drivers that drove both intended and unintended outcomes. Patterns of regularity were identified and synthesised to develop middle-range theories (MRTs). Results: Eleven MRTs demonstrate how and why intervention resources introduced in specific contextual conditions enact reasoning mechanisms and generate intended and unintended outcomes for patients, team members, the team and organisational leaders. The triggered mechanisms relate to shared mental models; openness, inclusivity and connectedness; leadership and engagement; social identity and intrinsic motivational factors. Conclusions: The findings provide valuable information for architects and facilitators of team interventions in acute hospital contexts, as well as help identify avenues for future research. Dataset: The data presented in this study are available on request from the corresponding author. The data are not publicly available due to their sensitive nature and potential identification of participants.

## 1. Introduction

The design and implementation of effective multidisciplinary team interventions to improve the quality and safety of care in acute hospital contexts are challenging tasks. Several factors can impact their effectiveness, for example questions over the validity of the interventions, staff competency and resources, lack of senior management support [1,2], hierarchies within teams [3], fragmented teams [4], staff turnover [5,6] and inability to take time out for improvement work.

For the purpose of this study, these interventions have been defined as:

“Two or more healthcare disciplines working together in an acute hospital context and in receipt of a programme or intervention or directly involved in implementation of a programme or intervention to improve team-working and/or quality and safety of patient care” [7,8].

The complex and dynamic hospital cultures and relationships, together with multi-layered structures, mean that these interventions are difficult to operationalise [9]. Each hospital context has its own unique culture, values and workplace relationships and rules, as well as its own requirements in terms of supporting these interventions [10]. Paradoxically, for environments that care for people, for staff hospitals are often highly charged places of work where there is increasing pressure to perform to targets and to prioritise between daily clinical, operational and administrative tasks. Finding and dedicating time to efforts to improve quality and safety can, therefore, be difficult. Whilst most staff accept that quality improvement is a necessary component of work [11], scheduling multiple staff members from different disciplines to attend at key decision-making moments, workshops or training programmes related to the intervention can be difficult to orchestrate and costly. It is important, therefore, that the interventions selected are appropriately considered, are efficient in their delivery and can be expected to be effective in terms of improving both patient and staff experience.

While there is evidence in the literature regarding the impacts of team interventions on team performance and patient outcomes, interventions that do not succeed are rarely reported or published. How and why some interventions fail whilst others flourish or how and why interventions may succeed in one context and fail in another are factors that are poorly understood. Studies that describe successful multidisciplinary team interventions usually provide a description of the goals and type of intervention, i.e., whether it was a change in practice, a process improvement or a training programme. This is then followed by the results, ascertaining the degree to which the intervention was a success or not and discussion about implications for future practice. The context in which the intervention is introduced, i.e., the enablers and contextual conditions of the intervention, are rarely described. Each hospital context is different and contextual variables can impact on the effectiveness of an intervention or the translation of an effective intervention from one hospital, specialty or department to another [12].

For example, there is increasing evidence to demonstrate that purposeful teaming and effective teamwork contribute to safer and better clinical practice [13,14]. In order to create the conditions for effective team performance and delivery of successful outcomes, architects or facilitators of team interventions should benefit from having an understanding of the conditions under which teams tend to enact certain types of co-ordination mechanisms to bring about successful outcomes [15].

In descriptions of team interventions, the active ingredients of the intervention that enact change are rarely described, assuming that it is the intervention as a whole that produces outcomes. There is little emphasis on generative causation and the evidence regarding mechanisms of action is only slowly accumulating [16]. Whilst behavioural change techniques such as the behavioural change wheel (BCW) [17] provide useful theory-based guidance and a structured method to facilitate development of interventions, “it is not possible to pre-determine which technique will work in which context” [18] (p. 18).

Recognising the gaps in the literature and the importance of contextual variables for translation of interventions to practice, we undertook a realist evaluation by exploring enablers and barriers to team interventions in acute hospital contexts [19]. We used this theory-based evaluation to ask what works for whom, in what conditions, why, to what extent and how?

Hospital multidisciplinary teams are complex and they operate within a complex healthcare system. Realist evaluations allow an appreciation of the fact that programmes operate within open systems with multiple factors interacting at different levels, producing both intended and unintended outcomes [20,21,22]. Realist evaluations are based on the premise that interventions are “theories incarnate” [9] and aim to unpack these underlying theories, which may cause changes in the form of “programme theories”. These theories are then refined through case examples, which help in understanding the mechanisms or in unpacking the ‘black box’ between intervention and outcome [23]. The goal is to produce more refined middle-range theories (i.e., “theories that have a common thread running through them that are traceable to more abstract analytic frameworks” [19]) of how the programme works. By understanding the contextual factors and the mechanisms through which outcomes are mediated, realist evaluators conclude that findings and recommendations are more relevant [24,25].

The stages of the realist evaluation process that evolved through this study have previously been outlined (see Figure 1).

### Research Aims and Objectives

A systematic search of the literature was undertaken using a realist synthesis approach [26] and interviews were conducted with practitioners in the field who had been involved in the delivery or design of interventions (key informants) [7]. Through these stages, we proposed 7 context–mechanism–outcome configurations (CMOCs) (see Table 1 below for explanation of realist definitions) that enable successful team interventions in acute hospital contexts and described how and why they work from the team’s perspective.

These initial programme theories (IPTs) were agreed to by a content expert advisory panel (*n* = 9) and are presented in the methods section of this paper in the form of if–then statements (see Table 2). In line with best practice for a realist evaluation [29], the purpose of this phase of the research was to test these IPTs using case studies. We chose to test the IPTs in two different hospital contexts in order to:Identify patterns showing what worked for whom, in which conditions, why, to what extent, and how;Use the evidence gleaned within the Irish case study and subsequently the US case study to support, refute or refine the IPTs;Synthesise findings from across the case studies to refine and abstract the IPTs to middle-range theories;Produce a set of general principles that will help to guide the implementation of multidisciplinary team interventions in hospitals.

A detailed protocol for this study was published in [8].

## 2. Methods

### 2.1. Ranking IPTs for Testing

The seven previously developed IPTs were first presented to the content expert advisory panel for ranking. This group (*n* = 9) comprised 2 international academic subject experts, 3 senior hospital managers, 2 practitioners in the field and 2 patient advocates. (Please refer to Appendix A). This panel was asked to rank the theories on a scale of 1 to 5 in terms how relevant they were in their experience in terms of how interventions work in order to reduce the IPTs to a manageable number for testing purposes. During this process, five of the seven IPTs were prioritised for testing, which are marked with asterisks in Table 2 below.

### 2.2. Selection of Case Studies

To enhance the rigour and robustness of testing, the content expert advisory panel agreed that two different team interventions from two different hospitals operating in two different health systems would be a requirement. Exploring two case studies in depth rather than multiple case studies at a more superficial level of analysis was deemed preferable in terms of extrapolating richer detail. Analysing more than two case studies was not possible within the timeframe of this study. Two cases were subsequently identified in line with criteria decided by the research team (see Appendix A).

Wong et al. [29] refer to the need to have multiple data sources to test theories. Being cognisant that the dynamic and adaptive local context could influence the intervention progression in unpredictable ways, prior to the data collection phase, the researcher developed a working knowledge and understanding of the broader hospital contexts in each case at the time of the intervention being completed. This was done both reflexively by reviewing minutes of meetings relating to the intervention and more pragmatically by developing field notes from conversations with key stakeholders in both hospitals.

***Case study 1****(**CS1**)* was a large quaternary academic teaching hospital in Ireland characterised by increasing demand for services without concomitant increased staff or bed capacity. Daily overcrowding in the emergency department (ED), high caseloads for already over-burdened “on-call” medical teams and mounting pressure from the national governing body to improve performance in unscheduled care created an urgency for change. There was a drive to reduce medical patients’ length of stay in order to increase bed capacity. There had been numerous attempts made to address this and there was a clear sense of intervention fatigue, scepticism and negative expectations at the time of embarking on the intervention.

*Intervention:* The team intervention was developed and facilitated by an in-house team with expertise in organisational change who employed the lean six sigma methodology [30] to re-design the process. Pre-intervention practice involved the designated “on-call” medical team taking over governance of medical patients who were admitted to the hospital through the emergency department. Whilst the acute medical specialty shared some of this workload during daytime hours, this still meant large numbers of patients who presented to the ED overnight required review by the “on-call” team the following morning. Post-call, there was, therefore, a large spike in caseload for the respective medical team. Requests for consults and subsequent takeover of care by other specialties was disorganised and time-consuming, often resulting in delays in the patient care pathway. The intervention sought to change this practice to a hospital-wide collaborative process of daily *takeover of care*. This resulted in a brief formal meeting each morning of all medical specialties where the “post-call” physician in consultation with colleagues handed over care to the most appropriate medical specialty. As the previous process had been in place for decades, this was a significant practice transformation.

*Team*: Following an invitation from facilitators to the general internal medicine (GIM) faculty seeking representation from each medical specialty, physicians (P) self-selected to participate in the GIM intervention team (*n* = 22). The facilitators specifically sought out the support of physicians who were either clinical leads for their respective specialty or those who appeared to be well respected by their colleagues. Facilitators referred to these physicians as *key influencers* in terms of their ability to promote positive engagement of their colleagues in the intervention. There was strong support from organisational leadership, with three senior managers (SM) included as core members of the intervention team.

*Methodology:* An intensive data collection phase (involving analysis of workload patterns across 14 post-call rounds and 308 patient pathways) was followed by a workshop to co-design a new way of working. This was subsequently followed by a series of monthly meetings to discuss plan–do–study–act (PDSA) cycles, review emerging data and make decisions based on progress. These were interspersed with smaller stakeholder engagement sessions as required. A new process was trialled and iterated over three PDSA cycles. This was followed by a six-month control phase once the intervention was embedded. The duration of the intervention was 15 months.

*Outcome:* As per local key performance data, there was a reduction in the length of stay for all medical specialties (range of 2–5 days) and improvements from baseline measures (range across specialties 22–40%) in terms of patients being governed by the appropriate specialty as per their primary diagnosis. Workload burden was more consistent and predictable and spikes in caseloads were avoided. The new process and way of working was successfully embedded and sustained over time.

***Case study 2****(**CS2**)* was a quaternary academic not-for-profit medical centre in the Pacific Northwest of the US. The advanced heart failure (AHF) faculty where the intervention was introduced was characterised by high turnover of nursing staff, low patient satisfaction, low staff satisfaction and high re-admission rates for patients. The AHF care teams had, therefore, been identified by organisational leadership as appropriate teams to engage in the intervention. There was a mixed reception for the intervention, with some staff being committed and engaged and others not as invested.

*Intervention:* The intervention was developed and facilitated by an academic practice partnership between a research team and the academic health centre care team. This team had expertise in IPCP, team science and quality improvement [31]. The focus of the intervention was to strengthen interprofessional collaborative practice (IPCP) by improving relational co-ordination, team communication and relationships [31,32,33]. Structured interprofessional bedside rounds (SIBRs) [34], which bring together health professionals using a structured format to collaboratively arrive at a daily plan of care, were developed and implemented and used as a vehicle to introduce IPCP. Pre-intervention, there was an ad hoc arrangement for rounding occurring in conference rooms or hallways, without the wider team members or patients present. SIBRs were, therefore, a significant change in practice.

*Team*: The purposefully selected change management team comprised multiple disciplines (physicians (P), nursing (RN/ARNP) and allied health professionals (AHP)) from across the faculty (*n* = 50). Organisational leadership support included attendance during project initiation and close out and at a celebratory workshop.

*Methodology:* Following a grant application process and formation of the change team, training in team strategies and tools to enhance performance (TeamSTEPPs) [35] for both the change team and the care teams took place over a one year period. This was followed by a longitudinal series of twelve leadership workshops delivered over a three-year period on a quarterly basis. The purposefully selected workshop topics included improving work and team processes, communication and relational co-ordination using a variety of evidence-based interventions, e.g., leadership coaching and presentations from field experts. The duration of the intervention was four years.

*Outcome:* Improved patient and staff satisfaction scores, reduced staff turnover and improved interprofessional collaborative practice were observed as key outcomes of the intervention [31,32,33]. SIBR has been successfully embedded and sustained over time is now considered the standard of care in these areas.

### 2.3. Testing IPTs

Both the content expert and methodology expert panel agreed that the IPTs should be tested and refined using data collected through theory-driven interviews with team members (CS1). A set of evaluative interviews had already been conducted on CS2, and given the period of time that had elapsed since the intervention, it was agreed that a realist analysis of this secondary data would be more appropriate than conducting a second set of interviews. Researchers’ insights could then be used to explore and unpick the underlying social and psychological drivers that drove both intended and un-intended intervention outcomes for team members using context–mechanism–outcome configurations as the units of analysis. By unpacking the generative causality, the IPTs could be examined in terms of their ability to explain what works for whom, in what context, to what extent, why and how. The ultimate objective was to develop a middle-range theory (MRT) that could be translated into a set of practical guidelines for programme designers and facilitators.

The testing of IPTs took place consecutively over a sixteen-month period (*CS1*: June 2019–February 2020; *CS2*: August–December 2020). The following is a summary of steps taken in this process (described in detail elsewhere [8]).

### 2.4. Data Collection: Case Study 1

As the primary researcher (UC) was involved in the design and delivery of this intervention, 19 interviews were conducted by a member of the research team who was unfamiliar with the intervention (EMcA). These were held in person. Manzano’s teacher–learner approach [36] was adapted to a more open style of interview in the initial stages to allow for participant led insights before moving to more theory driven questions. Please refer to Appendix A for an outline of the interview format. Interviews were audio-recorded. The mean interview time was 53 min (range 45–60 min).

### 2.5. Data Collection: Case Study 2

Data from 16 interview narratives (anonymised) obtained for primary research purposes were transferred to the primary researcher for the purposes of secondary analysis. These interviews had been conducted “to study participant reactions to, and perceptions of, workshop participation on personal leadership growth, its impact on team functioning and the overall IPCP environment” [31] (p. 77). These interviews were held in person. Please refer to Appendix A for an outline of the interview format that had been used in the primary study, which has been reported elsewhere [31]. The mean interview length was 43 min (range 30–60 min).

### 2.6. Data Analysis

The data underwent five phases of analysis by the primary researcher (UC), as depicted in Table 3 below. The approach to analysis was adopted from phases 3–5 of Gilmore’s realist evaluation analysis framework [37] (also see Appendix A for a detailed worked example of this process).

### 2.7. Data Preparation

The data from the audio files were transcribed (CS1) and uploaded (CS1 and CS2) to NVivo software [38] for transparency of analysis purposes. Each of the transcripts was read and annotations were made to note initial observations relating to the theories.

### 2.8. CMOC Extraction and Elicitation

Using deductive reasoning [39,40], all data relating to a programme theory were coded from each file to a corresponding adult node (i.e., a file that contained the aggregated data relating to respective IPT). Where new patterns emerged through inductive reasoning [37,39], data were coded to a new adult node labelled “new theories or insights”. Data relating specifically to contexts, mechanisms or outcomes that were new were coded under respective child nodes (i.e., a file that contains sub-coded data relating to the IPT). Where there was evidence of a new pattern emerging but still related to the programme theory, a new child node was created and named. Where there was evidence of two different patterns of generative causation (e.g., rival mechanisms) within a programme theory, theories were split into two further theories.

### 2.9. Using CMOCs to Refine IPTs

The respective narrative was analysed under each adult or child node. CMOCs were reviewed to determine how they aligned with the original IPT. This was done using a more advanced form of deductive and inductive reasoning referred to as “retroduction” [29], which involved moving back and forth within each file looking for patterns of regularity.

As per Gilmore’s guidelines [37], when a decision was made as to whether data supported, refuted or refined the IPT, the researcher made a note of how or why this decision was made. Interdependencies with other theories in the form of links and ripple effects were also annotated to support elucidation of patterns of generative causation. These data were then summarised under a linked memo to that programme theory.

### 2.10. Collating Evidence and Refinement Verification

Each of the memos specific to the five initial programme theories was read thoroughly. Patterns of regularity across files were noted. Where relevant, further iterative refinement occurred until it was evident that there was satisfactory evidence and explanatory power to support the refined theory. In some instances, decision-making required discussion with other members of the research team (A.D. and E.McA.).

Narratives frequently referred to the credibility of facilitators in CS1. Given that the primary researcher (UC) had been involved in delivery of this intervention, a decision was made to code these data under characteristics of the intervention that give credibility (i.e., IPT4). This IPT had not been selected for testing by the content expert group, but given the emphasis placed on this by participants in explaining the intervention’s success, it was agreed with the research team that it should be tested as part of the analysis. These data were independently and separately analysed by two authors to mitigate researcher bias.

It was also agreed that resource mechanisms should be disaggregated from reasoning mechanisms to provide more clarity [28] and that organisational contexts and team contexts should be explicitly identified.

For each case, a randomly chosen subsample of four files (20% of CS1 data 25% of CS2 data) was double-coded by another member of the research team (A.D.). The purpose of this was to challenge the primary researcher’s assumptions and interpretations, thereby enhancing the rigour and robustness of the process [40]. To enhance quality in the analysis of data from CS2 (where the primary researcher was less familiar with the intervention), the refined programme theories were presented to the team who facilitated the intervention (E.B., B.Z. and M.W.) for feedback and to challenge the inferences and assumptions made. This resulted in minor refinements.

### 2.11. Synthesis across Studies for MRTs

This final phase of analysis involved a search for demi-regularities (semi-predictable patterns of CMOCs) across the case studies. The primary researcher created a presentation to summarise each case outlining the programme theories from each case, demonstrating how the theories had evolved from the initial programme theory to the refined programme theories using coloured text. Using retroductive processes, which required movement back and forth between the two sets of findings and supporting files, the commonalities in the form of demi-regularities across the two case studies were identified. These findings in the form of middle-range theories (MRTs) were then presented to other members of the research team for further challenge until there was consensus.

MRTs are outlined below in the form of eleven “if–then” guiding principles. Whilst acknowledging that interventions are dynamic with multiple contextual conditions interacting at once and potentially changing over time, we used if–then statements to elucidate generative causation. The interdependencies between the MRTs are subsequently described.

We adopted a pragmatic approach to the reporting format. This aligned with the sequence of interventions, i.e., from pre-intervention planning to considerations during implementation of the intervention and factors relating to sustainability thereafter. By clearly identifying the resource mechanisms, i.e., the components “on offer” by the team intervention in each of the MRTs, this allowed us to identify the key mechanisms promoting specific outcomes. We used these resource mechanisms as titles for the MRTs.

## 3. Results

### 3.1. Sample Characteristics for CS1

Twenty-two individuals who attended meetings and workshops related to the intervention were invited to participate in an interview and 21 agreed to attend. Two were unable to attend due to scheduling challenges, resulting in a final sample of 19 participants (86 % participation rate). The 19 participants included physicians (*n* = 12), facilitators (*n* = 4) and senior managers (*n* = 3). Participants attended a range of 8–12 meetings. Of the interview participants, 10 physicians, 3 senior managers and 4 facilitators attended the co-design workshop.

### 3.2. Sample Characteristics for CS2

Fifty individuals attended meetings and workshops related to the intervention and those who attended 3 or more workshops were invited to participate in an interview. Twenty-four agreed to be interviewed. Seven were unable to attend due to scheduling challenges, resulting in a final sample of 16 (67% participation rate). The 16 participants included registered nurses (RN) (*n* = 7), physicians (*n* = 3), advanced registered nurse practitioners (*n* = 3), a social worker (*n* = 1) and an electrocardiogram technician (*n* = 1) (28).

### 3.3. Summary of Findings

A total of 27 demi-regularities emerged—fifteen from CS1 and twelve from CS2. Subsequently these were synthesised and abstracted into eleven MRTs. For a detailed example of how initial programme theories evolved to middle-range theories, please refer to Appendix A. The details of each of the 11 MRTS are first outlined below, followed by a focus on the interdependencies between these MRTs.

### 3.4. Middle-Range Theories (MRT)

Box 1MRT 1: Use of strategies to recruit team members and engage physicians.
*If*
There is purposeful consideration of team membership and recognition of the need for physicians to be engaged **Team context (Ct)**
*And the intervention*
Uses a range of strategies to invite members or to seek volunteers with deliberate engagement of physicians and key influencers **Resource mechanism (M Resource)**
*This enacts*
Feelings of being valued through recognition of need for diverse contributions, knowledge competency and new perspectives and recognition that physician engagement and support is key to success through empowering, motivating and engaging team members through a shared sense of ownership and purpose
**Reasoning mechanism (M *Reasoning*)**

*And results in*
More timely buy-in by other team members; more acceptable solutions with an increased chance of success; explicit acknowledgement and appreciation for others’ skills and contributions; evidence of willingness to share ownership of burden and wider participation of physicians through peer influence. **Outcome (O)**
*Evolved from IPT 5—Supported with minor refinement*
Evidence: *C1F1,C1P2, C1P3, C1P4, C1P5, C1P7, C1P8, C1P9, C1P12, C1P13, C1F3, C2RN2, C2P2, C2P3, C2RN4, C2RN3, C2AHP2, C2P4*

In both contexts, prior to the interventions, there was purposeful consideration of the team composition and a recognition of the need for physician engagement. Strategies were, therefore, used to recruit physicians, in particular those considered to be key influencers. Broader representation from other disciplines was not a feature in CS1 and team members felt this was appropriate, as the process being changed was operationalised by physicians. There was still broad representation within the medical division, however, as each of the specialties was invited to nominate a representative.

In CS2, the team comprised representatives from all disciplines and various levels of the organisation, including those who had a greater sense of urgency about the need for change in the working environment and were enthusiastic, invested, energised and committed.

In both cases, a range of strategies was used to invite members to participate, e.g., one-to-one stakeholder meetings or specific written invitations to physicians deemed to be key influencers in motivating for change.

In both cases, recognition of the value of individual team members’ contributions, knowledge and competency empowered, motivated and engaged team members, creating a shared sense of ownership and purpose. There was evidence of explicit acknowledgement and appreciation for others’ skills and contributions and evidence of willingness to share ownership of workload.

“I would say it’s … it was very empowering for the more junior faculty, who kind of felt like [they were] a little bit rudderless before this. …. It was very empowering for many of the APPs to make them feel part of the team. I … I’ve seen the engagement level of the nurses definitely increased with feeling part of that team”. C2P3

This also resulted in timelier buy-in by other team members and more acceptable solutions, with an increased chance of success. An unintended outcome as the interventions progressed was the wider participation of physicians through peer influence.

Box 2MRT 2: Clear and consistent communication and clarity of goals
*If*
There is recognition that there is underperformance or poor-quality care **(Co)** and there is a readiness to engage in change effort **(Ct)**
*And the intervention ensures*
Clear and consistent communication of goals, timelines and schedule of events; regular re-iteration and review of goals; effective use of data to evaluate progress; introduction of strategies and tools to communicate effectively and to ensure participation **(M *Resource*)**
*This enacts*
Shared understanding; shared situational awareness; clarity of roles and responsibilities; perceptions of confidence and trust in the intervention; motivation to question, understand and speak up
**(M *Reasoning*)**

*And results in*
Active engagement of the team members at meetings; use of a shared language; better relationships and team culture as evidenced by interactive behaviours; implementation of agreed solutions; speaking up and effective conflict resolution **(O)**
*Evolved from IPT2: Supported with minor refinements*
*Evidence*: CS1: C1F1, C1SM2, C1P3, C1P5, C1P7, C1P8, C1P10 C2RN2, C2ARNP1, C2P2, C2P3, C2RN3, C2RN4, C2AHP2, C2ARNP3, C2P4.

There was a recognition of underperformance or poor quality care at the organisational level and a readiness to engage in change efforts at the team level in both cases. Where the interventions enabled clear and consistent communication and clarity of goals, this supported the active engagement of team members, use of a shared language, better relationships and team culture through mechanisms including clarity of roles and responsibilities, perceptions of confidence and trust in the intervention and motivation to question, understand and speak up.

In each case, facilitators communicated timelines of events and schedules or re-iterated goals regularly, either at the outset of monthly team meetings or at the quarterly leadership workshops. The explicit statement of goals and distribution of performance data at timed intervals helped to maintain shared understanding and situational awareness throughout the intervention. Effective use of data to communicate progress and monitor performance was key to the engagement of physicians.

“So I think the data was key and at the end of the day somebody said afterwards that we’re all scientists in one way, shape or form. If you give us data you can show us up or move us along. So I think that was probably one of the more important things.” C1P13

Multiple communication tools and strategies were used to promote participation at meetings, motivating team members to question, understand and speak up. As per one team member:

“I think I’m getting better, as evidenced by this past weekend, at kind of just standing up and sticking my neck out and saying, “Oh, I really think that we should do X”, instead of kind of letting it pass by. Like this past weekend, the physician that I was on for rounds wasn’t necessarily an adopter of the process. But I just put my big girl pants on and I said, “Hey. We should go into the room and do rounds.” …. whereas before I just would’ve kind of let it slide and done the easy thing.” C2ARNP2

Box 3MRT 3: Use of strategies to stimulate interest and participation.
*If*
There are busy schedules and competing demands for the team and inter or intraprofessional tensions and rivalries **(Ct)**
*And the intervention*
Events are pre-planned in advance using thoughtful and engaging strategies (e.g., effective use of data, workshop content) to stimulate interest in attendance with a commitment to find suitable times to meet and/or staff are supported to attendKey champions and sponsors of the intervention are used as sound boards and supports(M Resource)This enacts:Self-interest, ability to focus, enjoyment, sense of fun, motivation and commitment to participate and a fear of missing out and inability to contribute if not present
**(M *Reasoning*)**

*And results in*
Increased attendance and ability to participate meaningfully in reflective practice, interactivity and acknowledgment of fun elements and building of inter- and intraprofessional relationships for team members.
**(O)**

*New theory*

*Evidence: C1F1, C1SM1, C1P1, C1F2, C1SM3, C1P13, C1F3, C2RN1, C2RN2, C2P1, C2ARNP1, C2P2, C2P3,C2RN4, C2AHP1, C2AHP2, C2ARNP3, C2P4*


A new programme theory was identified relating to pre-planning of strategies to effectively engage team members. In both case studies, facilitators were cognisant of busy schedules and competing demands and the existence of some inter- and intraprofessional tensions and rivalries. In these conditions, behind the scenes preparation for meetings (e.g., generating and disseminating data) emerged as a critical resource mechanism.

By preparing for meetings and using key influencers as sounding boards, facilitators created a safe discussion space where team members did not feel vulnerable or exposed, where they felt prepared and never expected to be surprised by data or emerging issues. Careful pre-planning and use of engaging strategies to stimulate interest enacted an ability to focus on the issues and created enjoyment, a sense of fun, motivation and commitment to participate. Physicians enjoyed sparring with colleagues and engaging in constructive debate “huffing and puffing” or bringing junior colleagues along to witness the “jibing” and “bit of fun” (CS1 File2). For some, this resulted in self-interest and a commitment to participate.

“I must say I never liked thinking that someone was planning how my service should work for me and not engaging me in it”. C1P8

For others, there was a fear of missing out on the ability to contribute if not present (C1F1, C1P6, C1P8, C1P10, C1F3).

There was evidence of self interest in team members’ own professional development, as well as self-interest in improving interprofessional collaborative practice via the intervention in CS2. These mechanisms resulted in well attended meetings despite many competing demands. In both studies, use of engaging strategies and time out from clinical activities resulted in meaningful participation in reflective practice.

“We were doing critical thinking. We were actually up there on the board. We were dialoguing. And I just thought that that was incredibly powerful, and then the way we shared those at the end I also thought was incredibly reflective and kind of set the stage for this shared knowledge about processes and the way we work as teams”. C2ARNP3

Box 4MRT 4: Co-designed approach.
*If*
The intervention team and facilitators are dedicated, enthusiastic and are collaborative in their approach **(Ct)**
*And the intervention*
is co-designed/decided by consensus; is patient-centred, relevant and problem-focused; is flexible in its application and the facilitators know when to engage or to step back and hand over ownership, using data effectively to communicate progress **(M *Resource*)**
*This enacts*
Self-awareness, mutual trust, shared understanding and ownership, giving a sense of credibility, logic, psychological safety and perception of being associated with something that might work as well as fear of missing out on ability to contribute if not present **(M *Reasoning*)**
*And results in*
Engagement with and progression of the intervention and implementation of a new co-designed approach to provision of patient care **(O)** 
*Evolved from IPT4 Refined*

*Evidence: C1F1, C1SM1, C1F2, C1P8, C1F3 and C2 FN C2P1, C2ARNP3*


In each case, the intervention team and facilitators’ dedication, enthusiasm and collaborative approach allowed for solutions to be co-designed, and this co-design element was particularly important to physicians, as they wanted to be involved in decisions that might impact them.

“There is a feeling that always one of our team would be there because you would be afraid that some decision might be made you know which would affect you so we would always, one of us would always try and represent us there”. C1P10

In addition, both interventions were grounded in reality and were patient-centred, relevant and problem-focused, as well as being flexible in their application. Facilitators also knew when to engage or to step back and hand over ownership.

“There were just so many areas that needed attention. And yet, as I tried to talk about those kinds of things, I really was perceived as an outsider. So the change really had to come from within the groups, and I had to kind of step back and observe the processes”. C2ARNP3

The co-designed approach and decision-making by consensus, together with the objective use of data, enacted self-awareness, mutual trust, shared understanding and ownership, giving a sense of credibility and logic, psychological safety, a perception of being associated with something that might work and a fear of missing out on the ability to contribute if not present.

“They felt they had ownership and they felt they could do something about it themselves and I think they felt this is ours …they never thought it was within their gift to give and make this change themselves. I think it did empower them a little bit … here they were very much brought on board with “you own this, you’re part of it, you deliver it”, they co-designed the solution”. C1P8

This resulted in engagement of team members and successful implementation of the new co-designed approach.

“People can resonate with it. It makes sense. And it was home-grown. That’s the best part”. C2RN4

Box 5MRT 5: Aligning with organisational goals and knowing when to leverage leadership support.
*If*
There is a strong driver for the intervention at the organisational level **Organisational context (Co)**, team goals are aligned with organisational goals and there is leadership support in the form of tangible resources **(Ct)**
*And where*
There are experienced facilitators who know the key moments when support of organisational leadership is required, there is regular appraisal and sharing of performance data with organisational leadership and acknowledgement and recognition for the team by organisational leadership when successes are noted
**(M *Resource*)**

*This*
Enlightens and interests leaders and motivates and empowers the team, giving a sense of influence, gravitas, validity and legitimacy to the intervention, reluctance to be perceived as inhibiting progress and connectedness and confidence in the process **(M *Reasoning*)** 
*This results in*
Evidence of team pride and camaraderie, easier implementation, team members demonstrating interest in sustainability and stating the potential to spread to other areas and/or to build on the success, public endorsement of the team’s work **(O)** 
*Evolved from IPT3 – Supported with refinement. New corollary theory also elicited*

*Evidence: C1F1, C1SM1, C1F2, C1SM3, C1P4, C1P5, C1P7, C1P10, C2P1, C2P3, C2RN3, C2AHP2, C2ARNP3, C2P4*


Team goals were aligned with organisational goals in both cases. There were also external drivers for the interventions through other national or state initiatives. The interventions were, therefore, seen as high priority for the respective hospital. It was important, therefore, that leadership was kept appraised of developments, and in both cases there were regular updates via e-mail correspondence and one-to-one meetings where performance data were shared. This approach kept leaders informed and interested.

The case studies differed, however, in terms of leadership support. Whilst both offered genuine support in the form of tangible resources, in CS1 the organisational leaders attended team meetings consistently and the visibility of the CEO at meetings was deemed an important factor in terms of participation.

“Because it made people realise the importance of the exercise. It’s not just a chat about or a moan about I’ve got too many patients to look after your patients, it was actually something that was very invested from the top down and I think it was, I think [name] being at those meetings, because he came to a lot of those meetings you know and I think it was, if you think about it, it was a really important move forward”. C1P4

When organisational leadership support was leveraged in both CS1 (e.g., securing recruitment of additional junior doctors) and CS2 (celebratory event), it motivated and empowered the teams giving a sense of influence and gravitas, validity and legitimacy to the intervention, a reluctance to be perceived as inhibiting progress and connectedness and confidence in the process.

“I think it assured them that this is going to happen, that the support was there and that this was you know not just being talked about and that you know it’s not going to look good for somebody to try and get out of this process”. C1P12

Box 6MRT 6: Visibility of leadership—corollary.
*If*
There is a lack of or inconsistent visibility of leadership at intervention events and/or inconsistent messaging from the team’s perspective **(Ct)**
*Despite intervention efforts*

*This enacts*
A perception of lack of shared ownership, feelings of disappointment and diminished motivation, lack of confidence and connectedness in the broader system **(M Reasoning)**
*And results in*
Less impactful or less effective team performance and slower implementation because of the impact inconsistent attendance of team members at intervention events and less physician engagement. Explicit statement of lack of organisational leadership support and impact on intervention. **(O)**
*New corollary theory to MRT 5*

*Evidence: C2RN2, C2P1, C2P2, C2ARNP2, C2RN3, C2RN5*


In contrast, in CS2 there was inconsistent attendance of organisational leadership at team events, which had its own impact from the team’s perspective.

“But I think we’ve always been missing that component of having some sort of higher leadership attending and really being a strong voice. As much as [name] and [name] want to do that, they didn’t attend to the change teams regularly enough or really make it their own”. C2ARNP2

The lack of visibility of leadership had a negative impact on team members enacting perceptions of lack of shared ownership, feelings of disappointment and diminished motivation, lack of confidence and connectedness in the broader system. Slower implementation progression due to inconsistent attendance of team members at intervention events and less physician engagement were issues reported by team members.

Box 7MRT 7: Creating opportunities to hear everyone’s voices and understand roles and responsibilities.
*If*
There is evidence of hierarchical structures or interprofessional/intraprofessional tensions and rivalries and a lack of understanding of others’ roles and responsibilities **(Ct)**
*And the intervention*
Creates opportunities for team members to be together in the same room, to develop relationships with each other and fosters an open and inclusive environment, where multiple methods are used for all team members to have an equal say and the need for interdependent work and relational co-ordination is made explicit **(M Resource)**
*This enacts:*
Building of personal connections; new insights and understanding of others’ roles and responsibilities; perception of being listened to and being supported; sense of an equal share, stake or ownership of the process; broader perspectives and mutual respect **(M Reasoning)**
*And results in:*
Staff reporting personal connections, more familiarity and less formality, more informal conversations and discussions, more staff speaking up and more collaborative practice,as evidenced by a team-based approach to care delivery and partnerships in delivery of patient care **(O)** 
*Evolved from IPT1 Partial support with some refinement*

*Evidence: C1F1, C1SM1, C1P1, C1SM2, C1F3 C2P1, C2ARNP1 C2P2, C2P3, C2ARNP2, C2RN3, C2AHP1, C2AHP2, C2P4*


There was evidence of hierarchical structures and interprofessional or intraprofessional tensions and rivalries and a lack of understanding of others’ roles and responsibilities (CS1 and CS2, field notes). The latter was more evident in CS2; however, it was reflective of the diverse professions that were involved. The interventions created opportunities for team members to be together in the same room and to develop relationships with each other. Multiple methods were used to enable team members to have an equal voice (e.g., liberating structures, critical conversations). This enacted an understanding of others’ roles and responsibilities, perceptions of being listened to and being supported, a sense of an equal share or ownership of the process, broader perspectives and mutual respect.

“And working with and hearing from all the multidisciplinary players in that room and kind of talking about their own obstacles or questions was interesting. Because I serve in an attending role, so it’s always nice to really understand what happens at various levels. Because you never really have the chance unless you query it. What are some of the problems or concerns they might be having? So it was good to kind of see how every player might potentially play in the whole process”. C2P2

Building of personal connections and gaining new insights featured strongly in both case studies. The daily processes introduced as part of the intervention offered opportunities to meet. During team sessions, there was more familiarity and less formality, more staff speaking up and evidence of more collaborative practice.

“It’s really all about relationship building … which sounds hokey, but boy is it true”.C2ARNP2

“Quite often people stayed around for quite a long time after [meetings] and chatted and caught up and you wouldn’t normally get to do that, definitely it was good for relationships and better collaboration”. C1SM3

This had a direct impact on patient care because of the new team-based approach to and partnerships in care delivery, resulting in shared plans of care (C1SM3, C2P2).

“There’s a couple of attendings who, when they’re on rounds, actively come and talk to me about, “Can you help me with this?” or “This was an issue,” or “This went really well,” and just, I think, see … it’s helped me gain with some of the attendings that I’m seen more as a partner when they’re on rounds with them, [than] someone who manages the unit, like someone who can really help the team and help the patients. So I think that has been a real positive”. C2P2

Box 8MRT 8: Building feedback loops to promote engagement.
*If*
There is endorsement of the intervention and an expectation that staff will participate with positive intent **(Co)**
*And*
If there is evidence of peers not buying into the process **(Ct)**
*And the intervention offers*
A mechanism for voicing concerns and escalation with rapid cycle quality and improvements in terms of corrective action **(M Resources)**
*This can*
Mitigate risk of feelings of being let down, a sense of disillusionment and a negative incentive for future participation **(M Reasoning)**
*And result in:*
Progression of the intervention towards successful outcomes or withdrawal from the intervention and poorer outcomes depending on the level of re-engagement of peers after the specific action to rectify the situation has been taken. ***(O)***
*New theory*

*Evidence C1F2, C1SM1, C1F2, C1P8, C1F3 also supported by C1FN, C2FN*


During the refinement verification and synthesis phases in CS2, it became apparent that when the interventions became “stuck” or met with opposition, there was a feedback loop to identify and address issues and move the intervention forward. This feature had not emerged previously during the development of initial programme theories and was subsequently verified by key informants for CS1.

As these team interventions were aligned with organisational objectives and were endorsed, there was an expectation that team members would engage with positive intent.

In these instances, each team was offered a mechanism for voicing concerns. In CS1, team members frequently approached the lead facilitator for this purpose. The issue would either be raised as an agenda item at the subsequent team meeting or of it was of a sensitive nature it may have been discussed in advance with either the project sponsors or one of the key influencers.

In CS2, opportunities existed to complete feedback forms if issues arose. In these instances, a respected senior leader in the team would try to intervene and move the issue along. This was helpful in terms of mitigating risk of feelings of being let down, a sense of disillusionment with the intervention and a negative incentive for future participation. As a result, there was either progression of the intervention towards successful outcomes in the respective unit or specialty or withdrawal from the intervention and poorer outcomes depending on the level of re-engagement of peers after the specific action to rectify the situation had been taken.

Box 9MRT 9: Expertise of facilitators.
*If*
There is observed dedication, tenacity and resilience of the intervention facilitators and a determination to keep going and adapting and flexing to changes in context over time **(Ct)**
*And the intervention*
Facilitators have expertise, are engaging and persuasive communicators and invest significant time and effort in the intervention **(M Resources)**
*This enacts*
Perceptions of credibility, recognition of investment and commitment, motivation, inspiration, respect and appreciation **(M Reasoning)**
*And results in*
Greater willingness to engage in change momentum and sustained engagement and progression of the intervention **(O)** 
*New theory*

*Evidence: C1F1, C1P1, C1F2, C1P2, C1P3, C1SM3, C1P6, C1P8, C1P9, C1P10, C1P11, C1P12*

*C2RN2, C2ARNP1, C2P3, C2ARNP2, C2AHP1, C2ARNP3, C2P4*


Team members for both interventions referred to the observed dedication, tenacity and resilience of the intervention facilitators (i.e., those who designed and delivered the intervention) and their determination to adapt to changes in context over time.

“And you guys sort of fortitude in saying, “We’re just going to keep doing this and trying to alter it as you did early on to make it shorter time periods so we could do it.” I mean it’s remarkable because it would have been very easy, I can only imagine, to get highly discouraged”. C2ARNP1

Facilitators were described as persuasive and engaging and their time and effort in the interventions enacted a perception of credibility leading to enhanced commitment, motivation, inspiration, respect and appreciation among team members. Many commented that without their dedication, progress would not have been made.

“We would still be going around in circles … without [name of facilitator] or something similar there’s no way … it wouldn’t have been achieved; I absolutely believe that”. C1P6

Box 10MRT 10: Supporting development of interpersonal relationships.
*If*
There is readiness for and openness to an improvement culture **(Co)**
*And the intervention offers*
Protected time and opportunities for the team to meet formally or informally
**(M Resource)**
Over time, a new context evolves, which supports the development of positive interpersonal relationships where there is increased familiarity and less formality among team members **(Ct)**
*This enacts*
Greater appreciation of and empathy for pressures on other team members, shared understanding of individuals’ skills and potential to contribute, a collective mindset, empowerment and a sense of psychological safety **(M reasoning)**
*And results in*
A positive team morale and working environment, ease of communication, openness and honesty, ability to progress intervention issues informally, pro-active helping behaviours or burden sharing, explicit statement of skillsets and preferences, conflict resolution and quicker recovery from conflicts **(O)** Evolved from IPT 6 Strongly supported—ripple impactEvidence: C1F1, C1SM1, C1P1, C1SM2, C1P3, C1SM3, C1P7, C1P13, C2RN2, C2ARNP2, C2RN4, C2RN7, C2AHP1, C2RN6

The development of interpersonal relationships was strongly supported as a key enabler for both interventions. In both hospitals, at an organisational level, there was readiness and openness to an improvement culture with both hospitals engaged in transformation programmes. Both interventions had been endorsed and were underway with both teams having protected time and opportunities to meet either formally or informally. Whilst there were a number of positive personal relationships in existence prior to the respective intervention either through prior experience of positive working relationships or already established social networks, the narratives from the case studies supported a more ripple impact from the teams having opportunities to meet both formally and informally specifically in relation to these two interventions. Having regular opportunities to meet throughout the interventions fostered the development of positive interpersonal relationships increased familiarity and facilitated more informal interactions among team members.

“I mean it brings people together doesn’t it, I think that’s a positivity in that and sometimes there can be little chats in there about other patients. Or you know if you are particularly worried about somebody even that you’ve seen in a clinic so I think the cohesiveness, the fact that it brings people together is a positive”. C1P2

These new conditions generated a greater appreciation of and empathy for pressures on other team members, empowered team members and developed collective mindsets. It also enacted shared understanding of individuals’ skills and potential to contribute and a sense of psychological safety.

“Having the workshops with the nurses, and with Teletech, and then getting to know them more on a social level. I think there’s a lot more interaction going both ways. Them feeling free to ask me more questions, or me feeling free to ask things of them or ask them questions … and so getting to know people on a personal level makes it a lot easier to work with people as a group, because you’re like, “Oh yeah, hey [name], I remember you……” Yeah, your guards are less up and, yeah, I think it’s just much more cool. And then, now that they’re joining us in rounds every day, I know who they are, what to expect of them, they know what to expect of me. So I think there’s a big improvement there”. C2ARNP2

These reasoning mechanisms resulted in openness and honesty, ability to progress intervention issues informally and effective conflict resolution. Other outcomes evidenced included quicker recovery from conflicts, enhanced communication, increased team morale, a more positive working environment with evidence of pro-active helping behaviours and burden-sharing.

Box 11MRT 11: Celebrating and building on success.
*If*
There is acknowledgment of success of the intervention and more positive working relationships **(Ct)**
*And the interventions allows for*
Demonstration and acknowledgement of success **(M Resources)**
*This enacts*
A sense of personal contribution, connection with something positive, team members aligning with this success and a boost to team morale or a feel good factor **(M Reasoning)**
*And results in*
Evidence of camaraderie; a new way of working for the team, new staff adapting to this as embedded part of the culture; externally perceived credibility in the intervention and subsequent buy in from other staff; sustainability of the intervention and potential to spread; participants’ willingness to engage in other interventions **(O)** *Evolved from IPT 4a* Supported 
*Evidence: C1SM2, C1P3, C1P12, C1P13, C1F3, C2RN1, C2RN2, C2P1, C2P2, C2ARNP2, C2RN4, C2RN5, C2AHP1, C2AHP2, C2P4*


In each case, when teams saw evidence of improvements and these were acknowledged as successes, this helped maintain the momentum.

“Because like anything if you get a sense that something is successful in any small way then that will accelerate, that’s almost starting in the middle of the story”. C1P12

Where the intervention was acknowledged (e.g., via hospital newsletters, at organisation-wide presentations) and successes were celebrated, this enacted a sense of personal contribution and connection with something positive. Team members seemed to want to align with this success, boosting team morale, creating a ‘feel good’ factor which resulted in team camaraderie. New ways of working were also embedded and sustained. There was externally perceived credibility for both interventions and buy-in from other staff. In both cases, team members openly expressed a willingness to engage in other interventions, while in CS1, a number of team members saw it as a critical opportunity to build support and momentum for other improvements or developments. (C1SM1, C1P1, C1P13).

### 3.5. Interdependencies

We have presented eleven middle-range theories in order to reflect the semi-predictable patterns of occurrence across the two case study contexts. There is an inherent risk in these being perceived as linear. Whilst we felt it was important to illustrate the MRTs in order to depict the generative causation, below we use three examples of interconnectedness between MRTs in recognition of the patterns of interdependencies.

### 3.6. Foundational Logistics

In both cases, there was purposeful selection of team members with deliberate physician engagement (MRT 1) and pre-planning of meetings, which took into account busy schedules and competing demands (MRT 3). Together these created the opportunities for diverse team members to come together in the same room and allowed their voices to be heard. This openness and inclusivity were key enablers for better understanding of roles and responsibilities (MRT 7). This was also enhanced by the team members’ willingness to engage in change effort (MRT 2) and facilitators’ expertise in engaging and persuading team members to attend meetings and workshops (MRT 9). The introduction of strategies and tools to communicate effectively (MRT2) then helped to motivate these team members to question, understand and speak up. The interplay between resource mechanisms therefore strengthened the enabling conditions, i.e., for diverse team members to come together and engage in meaningful and constructive dialogue, moving the interventions forward within a difficult context of interprofessional or specialty tensions and rivalries and traditional hierarchies.

### 3.7. Expert Facilitation during the Intervention

There were strong interdependencies between the expertise of the facilitators (MRT 9) and a number of other resource mechanisms across both case studies. Co-design was feasible because there was a willingness of team members to engage in the process (MRT 4); engagement in the process was enhanced as there was trust and credibility in the intervention, which was enabled by experienced facilitators who communicated clearly and consistently (MRT 2). Organisational leadership support was also seen as key to motivating and empowering team members to be involved in the intervention (MRT 5). In both cases, the facilitators were instrumental in leveraging this support at key moments. By facilitators keeping organisational leadership enlightened and interested in the intervention through regular appraisal and effective use of data (MRT 2), there was evidence of organisational leadership commitment in the form of resources and acknowledgement of successes, which resulted in easier implementation of the intervention.

### 3.8. Sustainability Factors

Supporting development of interpersonal relationships (MRT 10) was a key contextual condition for sustainability of the intervention in the longer term. This was enabled by opportunities for everyone to be in the same room together in an environment that was open and inclusive (MRT 7). This was only made possible, however, because of the ‘protected time’, which was enabled because of pre-planning of meeting and workshop schedules, a commitment to find suitable times to meet and staff being supported to attend (MRT 3). It was the interplay between these resource mechanisms that helped to foster the development of personal connections and increase familiarity and reduce formality between team members, resulting in more positive working relationships. Consequently, this had a ripple effect. Team members were able to celebrate and build on this success, leading to further strengthening of positive relationships, continued engagement in the process and a new way or working (MRT 11).

## 4. Discussion

Given the high-risk and complex nature of acute hospital contexts, effective and efficient team interventions are required more than ever to help mitigate risks and to optimise quality and safety of patient care [41]. The purpose of this research was to test five IPTs relating to team interventions in acute hospital contexts across two diverse contexts in order to understand what works for whom, in what conditions, why, to what extent and how? Despite the two contexts being complex and diverse, strong patterns of regularity emerged in terms of how the resource mechanisms impacted team members’ reasoning and generated resultant outcomes. We synthesised these patterns of regularity to elucidate 11 middle-range theories. We have also outlined the interdependencies between these MRTs. As a final plausibility check, these MRTs were substantiated and contextualised within the broader context of the theoretical and empirical literature [42].

### 4.1. Shared Mental Models

As in many hospitals, in both case study contexts, there was a recognition of inefficiencies and poor quality care and a readiness for change. Staff had multiple competing demands for their time. Our findings illustrate that in these conditions, shared mental models become essential to team functioning, relating positively to team process and performance. The importance of shared mental models in team performance has previously been cited [41,43,44,45,46]. Under stressful conditions, it is difficult for teams to engage in strategizing [43]. In these conditions, clear and consistent communication and clarity of goals ensures team members have a shared sense of ownership and trust, clarity of role and purpose and shared situational awareness (MRT2).These mechanisms are important for team building and team performance [47,48,49].We demonstrate that through these mechanisms, effective communication ensures that team members know what is required of them and their team, enabling meaningful engagement and participation in the intervention, improved relationships and team culture, effective decision-making and implementation of agreed solutions (MRT 2).

### 4.2. Openness, Inclusivity and Connectedness

In both cases, experienced facilitators fostered open and inclusive environments using multiple strategies to ensure there was safe space for all voices to be heard (MRT 7). This enacted a sense of psychological safety, allowing team members to become more familiar with one another, interactions to become less formal and interpersonal relationships to be developed (MRT 10). This is significant in a context where inter- or intraprofessional tensions and rivalries exist. The positive impacts of openness, inclusivity and connectedness and their relationships with better team performance [50], team resilience [51], relational dynamics [50,52] and relational co-ordination [6] are well documented. Mazmanian and Perlow [53] recognised respect, openness and connectedness as important concepts for effective team performance and called for the use of “spaces and interaction scripts” to foster these characteristics amongst team members (*p*.118). As per our findings, *safe spaces,* which support the development of interpersonal relationships and allow staff to engage in reflective practice, are important in order to develop a greater appreciation of, and empathy for, pressures on other disciplines, as well as new insights into and understanding of other peoples’ roles and responsibilities (MRT 3, MRT 7, MRT 10). These conditions broaden team perspectives and enact a sense of mutual respect and value. In acute hospital practice, with ever increasing daily pressures, finding space and “time out” for teams to just be together for formal and informal gatherings can still be a challenge [41]. As our study demonstrates (MRT 3), this is an important practical consideration to enable effective practice transformation and one that should not be ignored or de-prioritised.

### 4.3. Leadership and Engagement

Physician engagement and senior management support are cited as important enablers for interventions to improve quality and safety of care [54,55,56]. Our research findings explain how organisational leadership support enacts a perception of influence and gravitas and also gives a sense of legitimacy and validity—factors that are especially important for sceptics of the intervention [55]. Findings demonstrate that organisational leadership is empowering and motivating for team members and that there is a reluctance to be perceived as inhibiting progress when leadership is visible. Organisational leadership support also engenders connectedness and confidence in the intervention process (MRT 5). When organisational leaders take into account staff priorities and teams align their interventions with strategic objectives of the organisation, interventions are likely to have a better chance of success [56,57,58].

For broader team membership, there is a recognition that physician engagement is also key to success because it creates a shared sense of ownership and purpose and results in more timely buy-in of other staff, with evidence of willingness to share ownership of burden across team members (MRT 1). As outlined by Ling [59] and Woods et al. [55], physician engagement can depend on the credibility of the intervention, ensuring that there is empirical evidence to support it and implementers who are well briefed and able to confront challenges. We also found support for this.

For physicians, the co-design of content that is patient-centred, relevant and problem-focused helps to enact credibility in the intervention and gives a perception of being associated with something that might work or a fear of missing out and inability to contribute if not participating in the intervention (MRT 4). Unsurprisingly perhaps, co-design approaches are increasingly being used effectively for quality improvements in healthcare [47,49,60,61].

For team members in this study, the co-design approach enacted a shared sense of ownership, a sense of credibility and logic and a sense of psychological safety. As per MRT 9, expert, persuasive and engaging facilitators enact a sense of credibility through their dedication, tenacity, resilience and ability to flex and adapt to changes. This motivates and inspires team members resulting in a greater willingness to engage in change momentum.

### 4.4. Social Identity

Our findings suggest that team members identify strongly with the success of interventions. The acknowledgement of success and dissemination of the success story are important features in terms of sustainability, resulting in increased camaraderie, ‘buy-in’ from other staff and a willingness to continue with the intervention and build on its success. The importance of celebrating success is frequently cited in change management literature [62,63]. The findings in this study demonstrated that celebration of success enacts a sense of personal contribution, a connection with something positive and a desire to align with this success, boosting team morale and engendering a ‘feel good’ factor (MRT 11).

These findings are further substantiated by social identity theory literature. As theorists in this domain posit, people seek out positively valued traits, attitudes and behaviours that can be seen as characteristic of their in-groups. Group membership helps people to define who they are and to determine how they relate to others [64,65,66]. In particular in CS2, as the intervention progressed, peoples’ impression of their team’s identity changed and there was a greater appreciation of roles and skills of others. Having a strong team identity can help reduce silos within teams [67,68]. It is important to team members to maintain a positive image of the team to which they belong [69]. The need to celebrate is reasoned by team members to be an important overt expression of the team’s success and is also reflective of their own personal positive contribution (MRT 11).

### 4.5. Motivation

Team interventions must often rely on the intrinsic motivation of hospital staff to maximise the quality and safety of care they provide for patients [55]; however, motivating factors may not always be entirely altruistic, as per our research findings. When organisational leaders endorse and support interventions, there is a sense of expectation or “what’s in it for me”? Expectancy theory [70] is often used to explain employee motivation as “the degree to which an effort is perceived to lead to performance, performance leads to rewards and the rewards offered are desirable” [71]. Extrinsic motivators (e.g., an expectation of resources (CS1) or an opportunity for professional development (CS2)) may help to explain why team members behave in a certain way. Team members may be motivated and committed to participate partly because of self-interest (MRT 3). Where practice change impacts on team members’ daily routines directly, this is understandable and partly attributes for why co-design principles work. Understanding why team members want, intend or decide to participate in an intervention is important. Their expectations may need to be fulfilled or managed to motivate them to meaningfully engage. We are cognisant that all team members do not respond in a similar way to the resource mechanisms offered by an intervention [28]. The degree to which motivation is activated is likely to differ for individual team members.

### 4.6. Strengths and Limitations

Several authors have tried to establish which aspects of complex interventions are important in terms of contributing to effectiveness; however, difficulties arise in trying to generalise findings because of lack of methodological rigour and objectivity of data [33] or because of the level of complexity of the interventions and consequent inability to identify the active ingredients of the intervention that impacted the outcomes [72].

In contrast, realist evaluation allows an appreciation of the fact that programmes operate in open systems with multiple factors interacting at different levels, producing both intended and unintended outcomes [20,21,22].

The IPTs were tested across two diverse contexts. This adds significant strength to the findings. In this study, we not only unpacked the resource mechanisms of the intervention that enacted change, we also unpacked how and why this was the case and identified under which contextual conditions the mechanisms worked. The context-specific detail that we were able to extrapolate ensures a richness to the quality of the data. The subsequent abstraction to produce eleven middle-range theories in the form of generic principles allows for transferability and scalability of these principles to support team interventions in other hospital contexts [24,25], overcoming the challenges previously identified in the literature. By disaggregating the resource mechanisms from the reasoning mechanisms, the key resources of the intervention that triggered reasoning to enable specific outcomes were identified. These resource mechanisms have important practical application for facilitators or other implementers of interventions in terms of how they design and implement interventions. The reasoning mechanisms identified help to deepen our understanding of how and why resources introduced under specific contextual conditions are likely to bring about outcomes.

Some challenges were experienced in terms of data collection. The teacher–learner approach [36] used in CS1 required adaptation to encourage participant insights. For this reason, the interviewer adapted the interview guide to be theory-driven, although in the initial stage of the interview a more open approach was adopted, allowing new (participant-led) theories to emerge.

Given the time that had elapsed since the intervention in CS2, a decision was made to rely on secondary data from a primary dataset. This meant the loss of certain details that may have been elicited specific to the IPTs if purposeful interviewing had been completed; however, the secondary analysis of transcripts facilitated to a more rigorous testing environment, i.e., the interview guide, was not influenced by the researchers’ theories. This further bolsters the credibility and trustworthiness of the findings.

### 4.7. Future Research

This study elicited important detail in terms of what works for team members in implementing team interventions in acute hospital contexts. While the MRTs offer a set of generic principles for facilitators of team interventions and for those involved in quality improvement work, we recommend that they are adapted and disseminated for practical use by facilitators and implementers of interventions. Their utility and usefulness in practice can then be evaluated across a broader number of hospital and healthcare contexts.

## 5. Conclusions

This study details how we tested 5 IPTs across two diverse hospital contexts in order to glean important information about enablers and barriers to effective team interventions in acute hospital contexts. The findings are presented in the form of MRTs illustrating how and why the resource mechanisms work in specific contextual conditions, for whom they work and why.

This study makes an important contribution to the literature in identifying the resource mechanisms, i.e., the active ingredients of the intervention that enact change in specific contextual conditions through the reasoning mechanism that they trigger. In describing the generative causation and mechanisms of action, we highlight how and why mechanisms relating to shared mental models; openness, inclusivity and connectedness; leadership and engagement; social identity and intrinsic motivational factors bring about outcomes for patients, individual team members, the team as a collective and organisational leaders. Our insights, therefore, offer valuable information for architects and facilitators of team interventions in acute hospital contexts.

We recommend that the MRTs are adapted for practical use so that their usefulness in helping to deliver more effective team interventions can be measured in terms of the impacts on quality and safety in hospitals.

## Figures and Tables

**Figure 1 ijerph-18-08604-f001:**
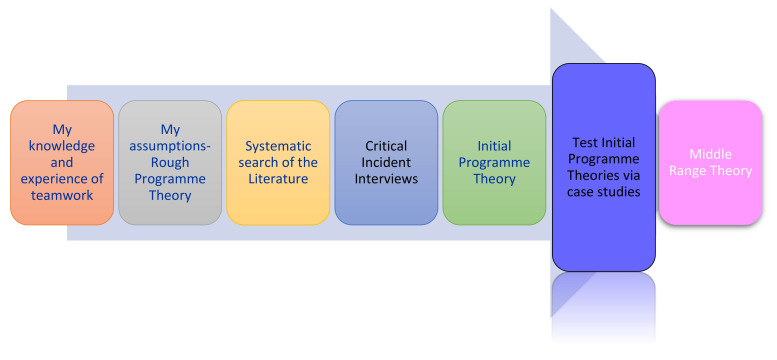
Stages of the realist evaluation [7,8].

**Table 1 ijerph-18-08604-t001:** Realist definitions.

	Definition
**Context**	Those features of the situation into which programmes are introduced that affect the operation of programme mechanisms [27] Organisational context (*Co)—features of the hospital Team context (*Ct)—features of the team
**Mechanism**	A combination of resources offered and the participants’ reasoning in response Resource—the component introduced in a context Reasoning—the way in which the participant interprets and acts upon the resources introduced as part of the intervention [28]
**Outcome**	The intended and unintended consequences of the intervention.
**Configuration**	Patterns and variations in patterns (*CMOC)
**Demi-regularity**	Semi-predictable pattern of occurrences within the data
**Initial Programme Theory**	The programme architect’s articulation of how the intervention is expected to lead to its effects and in which conditions it should do so
**Middle-Range Theory**	“Theories that have a common thread running through them that are traceable to more abstract analytic frameworks” [19]

CMOC, context–mechanism–outcome configuration; Co, organisational context; Ct, team context.

**Table 2 ijerph-18-08604-t002:** Initial programme theories (IPTs).

CMOC	Context	+Mechanism	=Outcome
1 * Interdisciplinary team approach and flattened hierarchy	*If* Each team member’s voice is heard and considered of equal value	*Then this enacts:* understanding of roles, mutual respect, support and value; self and team efficacy; perception of shared decision-making and common purpose	*Resulting in:* increased job satisfaction; higher levels of competence; better teamwork; lower feelings of emotional exhaustion; breakdown of interprofessional silos; more integrated care; connectivity of the team and camaraderie; more efficient use of time
2 * Effective communication and shared understanding of goals	*If* there is clear, simple, open, honest and timely communication in an appropriate and inclusive environment with specific, measurable, achievable, realistic and time-bound (SMART) goal-setting	*Then this enacts:* shared understanding and clarity of role and purpose; self-worth and value; perceptions of confidence and trust in the intervention	*Resulting in:* positive engagement of the team; situational awareness; more integrated planning; more efficient use of time; better chance of success
3 * Leadership support and alignment of team goals with organisational goals	*If* there is genuine leadership support in the form of tangible resources, positive acknowledgement of staff and alignment of team goals with organisational goals through effective engagement and dialogue	*Then this:* motivates, empowers and engages staff; enacts a sense of team efficacy; enacts a perception of the intervention making sense and a shared sense of responsibility and accountability	*Resulting in:* team pride and camaraderie; connectedness and confidence in the broader system; easier implementation and sustainability of the intervention
4 Characteristics of intervention that give credibility	*If* the intervention is facilitated or driven by experienced facilitators who staff can relate to and trust, with appropriate clinician involvement where relevant, and with perceived relevance to practice with clearly defined goals/outcomes	*Then this enacts:* team pride and camaraderie; connectedness and confidence in the broader system; easier implementation and maintenance of the intervention	*Resulting in:* team pride and camaraderie; connectedness and confidence in the broader system; easier implementation and sustainability of the intervention
4a *Ripple theory* Evidence, recognition and celebration of success	*If* there is evidence of a positive outcome and there is recognition and acknowledgement that an intervention is successful	*Then this:* empowers, motivates and incentivises staff	*Resulting in:* externally perceived credibility in the intervention and subsequent *buy-in* with increased likelihood of further engagement and spread of the intervention and/or future team interventions
5 * Appropriate team composition and physician engagement and support	*If* there is broad and purposeful selection of team members *with* physician engagement and support if intervention has a clinical focus	*Then this enacts:* feelings of knowledge, confidence and competency; psychological safety; perception of power and influence	*Resulting in:* legitimacy of the intervention; better and more timely “buy-in”; staff satisfaction; translation of intervention outcomes to practice and better chance of sustainability
6 * Personal relationships	*If* team members have positive personal relationships or prior experience of a positive working relationship and/or an established social network	*Then this enacts:* perceptions of trust; perceptions of psychological safety; shared understanding of experiential knowledge of team, including ways of working, skillsets, likes and dislikes	*Resulting in:* better engagement in intervention and easier implementation; ability to progress intervention issues informally; distribution of work according to skillsets; more honest and open communication; more integrated planning; quicker recovery from conflicts
7 Interprofessional tensions	*If* there are interprofessional tensions, rivalry and mistrust	*Then this enacts:* feelings of frustration; lack of respect; dis-empowerment, perceptions of lack of psychological safety and cynicism	*Resulting in:* failure to progress the intervention; lack of support for the intervention and/or withdrawal from the process
7a *Ripple theory* Escalating mechanisms	*If* there is a failure to progress an intervention, lack of support for the intervention and/or withdrawal from the process because of interprofessional tensions	*Then this enacts:* further escalating mechanisms of dis-satisfaction; depletion of energy and resilience; perception of powerlessness	*Resulting in:* greater silo mentality among professions

Reproduced with permission from [8]. * Refers to those IPTs that were ranked for testing.

**Table 3 ijerph-18-08604-t003:** Data analysis phases.

Data Analysis and Synthesis within Realist Evaluation (27)		
**Phase 3**	**Step 1: Data preparation**	✓ Data from the audio files transcribed *(CS1)* and uploaded *(CS1 and CS2)* to NVivo software (28) ✓ Transcripts read and annotations made
	**Step 2: CMOC extraction and elicitation**	✓ Using deductive reasoning and inductive reasoning—data coded to adult nodes 6.0 or new child node CMOCs were extracted and/or new CMOC elicited
**Phase 4**	**Step 1: Using CMOCs to refine IPTs**	✓ Using retroduction- respective narrative analysed under each adult or child node. ✓ CMOCs reviewed to determine how they aligned with the original IPT.
	**Step 2: Collating evidence and refinement verification**	✓ Memos specific to the five individual programme theories read thoroughly Patterns of regularity across files noted. Thought processes annotated. ✓ Decisions were logged. ✓ Supporting evidence for thought processes logged with examples of supporting quotes. ✓ Randomly chosen sub sample of four interviews was double coded from each case study (*n* = 8).
**Phase 5**	**Step 1: Synthesis across studies for MRTs**	✓ Narratives searched for demi-regularities (semi-predictable patterns of CMOCs) across case studies.

(Methodology expert panel—a group of researchers with an interest in realist methods, *n* = 8).

## Data Availability

Datasets available on request form corresponding author only as the data are sensitive and participants may be potentially identifiable.

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
