# Peer review of "A Realist Evaluation of Team Interventions in Acute Hospital Contexts—Use of Two Case Studies to Test Initial Programme Theories"

_ijerph, 2021, doi:10.3390/ijerph18168604_

Round 1

Reviewer 1 Report

In the introduction section authors say that the action mechanisms of the interventions are rarely mentioned, so they should have reviewed what has been published about the Behavior Change Techniques developed by Dr. Susan Michie et. al from University College London. Several interventions have been designed using these techniques. There are BCTs to carry out interventions related to different types of behavior and it is my understanding that in this study the change in administrative processes is proposed. I suggest that authors should mention these techniques and even to differentiate between them and the one proposed in this study.

· The composition of the group of experts is mentioned in the introduction, while in the methods section is not mentioned.

· The reference to the group of experts is wrong, since it is mentioned that it is file 2 of the supplementary material. However, when reviewing the material, the conformation of the group of experts is found in supplementary file 1.

· The composition of the group of experts in terms of their profile is attached. however, the number of people who participated as experts is not specified, so this should be mentioned in the article.

· In the article practitioners are mentioned, while in the supplementary material users are mentioned.

· In the supplementary material in file 3, authors jump from 3 to 6.

· In the article, there is a reference to file 2 of the supplementary material, which is made twice. One refers to the composition of the group of experts, while the other details the case studies.

· Please detail how interviews with participants were held in the two case studies. In the manuscript authors mention that for more information about interviews reference 8 should be consulted (Cunningham U, De Brún A, Willgerodt M, Blakeney E, McAuliffe E. Team interventions in acute hospital contexts: protocol for the evaluation of an initial program theory using realist methods. HRB Open Res. 2021 Mar 29; 4:32.). When one consults this reference, in turn it claims that for more details about the interviews reference 41 should be consulted (Cunningham U, De Brún A, Mayumi W, et al. : Appendices interview formats. Dryad, Dataset, 2021. http://www.doi.org/10.5061/dryad.q83bk3jg8). Although references are included, it is necessary to briefly describe whether the interviews in the two cases were held in person or virtually.

Author Response

Kind regards

Una

Reviewer 2 Report

Thank you very much for the opportunity to review this manuscript! Overall, I would suggest making the manuscript more concise and avoiding repeating ideas – this would help them stand out rather than get lost in the detail.

Please see my comments below:

1) Abstract: the sentence in the Results appears too long, consider breaking it up in 2 sentences

2)  Figure 1 – please spell out IPT, even though it is spelled out in the text

3) Table 1: abbreviations (CMOMC, Co, Ct) – are unclear, please spell them out in the note beneath the table (even though you spelled them out in the text

4) Table 2: please spell out SMART goal – while this is a commonly used term, it is still best practice to spell out all abbreviations

6) Line 170: unclear what is meant by “part-admitted”

7) Line 173: you may delete the word “unnecessary” – delays are usually unnecessary

8) Methods: who developed the interventions that were implemented in the 2 hospitals?

9)  Lines 179-194: how many people were on the team?

10) Lines 181-182: unclear what is meant by this, vague sentence – who were the key influencers and what did deliberate engagement with them presume?

11) Line 190: please spell out PDSA the first time the abbreviation is used

12) Line 196-197: unclear what is meant by improvement – this needs to be reported in a percentage – or how did you realize that there was improvement in precise match between patients’ primary diagnosis and their assigned medical specialty?

13)  Line 222: please spell out TeamSTEPPS the first time it is used

14) Lines 255-256, 272-273: typically, average interview length is reported with the range (e.g., mean interview length was X minutes, range (Y-Z)).

15) Line 274: when a sentence starts with a number, please spell it out

16) Lines 275, 277: please round the number to 67%

17) Table 3: phase 5 – how many interviews were in the 4 files?

18) Table 3: unclear why certain sub-headings are bolded

19) Data analysis: how many investigators analyzed data?

20) Data analysis: please provide the definition of “adult node” and “child node”

21) Line 295: no need to put quotation marks around the word “new”

22) Lines 351-353: this is a long sentence, with the use of “in terms of” twice, please consider re-phrasing into 2 shorter sentences

23) Lines 370-371 – you continue to refer to key influences, please define this term earlier in the paper – it appears intuitive, but it would be helpful to know who these are – e.g., those in administrative positions, those who have the longest work experience in this hospital, those who are chairs of departments, etc.?

24) Lines 392-394 – unclear what is meant by “broadening engagement of physicians though peer-led influence.”

25) Line 414: comma is necessary before “motivating”

26) Line 438: it looks like the abbreviation should have been C1P8 – not CIP8

27)   MRT9 – who were the facilitators? E.g., were these the physicians/other healthcare professionals who assumed leadership for the interventions?

28) Lines 602-605 – this sentence seems repetitive to what was already stated in the paragraph

29) Lines 648-649: no need to state “that are in evidence.” Throughout the entire paper – please revise to delete unnecessary words and make the text more concise. It reads more like a textbook chapter than a concise article. I would advise highlighting the most important aspects of the results and the discussion, and shortening non-essential details and repetitions as much as possible

30) In the beginning of the paper, please provide a definition of a middle-range theory

31) Line 677 – “when needed” is unnecessary

32) Line 681 – “being created” is unnecessary

33) Line 693 – “ever increasing complexities across multiple boundaries” – unclear what is meant here

34) There is not enough discussion of unintended consequences of these interventions, the authors appears to focus predominantly on successes

Author Response

Kind regards

Una

Round 2

Reviewer 1 Report

No comment, my comments were addressed